# Awareness, treatment, and control among adults living with arterial hypertension or diabetes mellitus in two rural districts in Lesotho

**Lucia González Fernández**[1,2,3], **Emmanuel Firima**[1,2], **Ravi Gupta**[4], **Mamoronts'ane Pauline Sematle**[4], **Makhebe Khomolishoele**[4], **Manthabiseng Molulela**[4], **Matumaole Bane**[4], **Mosa Tlahali**[5], **Stephen McCrosky**[1], **Tristan Lee**[1], **Frédérique Chammartin**[1], **Eleonora Seelig**[6], **Felix Gerber**[1,2], **Thabo Ishmael Lejone**[1], **Irene Ayakaka**[4], **Niklaus Daniel Labhardt**[1‡], **Alain Amstutz**[1,7,8‡]*

1 Division of Clinical Epidemiology, Department of Clinical Research, University Hospital Basel, University of Basel, Basel, Switzerland, 2 Department of Medicine, Swiss Tropical and Public Institute, Basel, Switzerland, 3 SolidarMed, Partnerships for Health, Luzern, Switzerland, 4 SolidarMed, Partnerships for Health, Maseru, Lesotho, 5 Mokhotlong District Health Management Team, Ministry of Health Lesotho, Mokhotlong, Lesotho, 6 Department of Endocrinology, Diabetes and Metabolism, University Hospital Basel, Basel, Switzerland, 7 Oslo Center for Biostatistics and Epidemiology, Oslo University Hospital, Oslo, Norway, 8 Population Health Sciences, Bristol Medical School, University of Bristol, Bristol, United Kingdom

‡ NDL and AM share last authorship on this work.
* alain.amstutz@unibas.ch

**Data Availability Statement:** We deposited an anonymized dataset with the data presented in the

## Abstract

In Lesotho, the hypertension and diabetes care cascades are unknown. We measured awareness, treatment, and control of hypertension and diabetes among adults ≥18 years and identified factors associated with each step of the cascade, based on data from a population-based, cross-sectional survey in 120 randomly sampled clusters in the districts of Butha-Buthe and Mokhotlong from 1st November 2021 to 31st August 2022. We used multivariable logistic regression to assess associations. Among participants with hypertension, 69.7% (95%CI, 67.2–72.2%, 909/1305) were aware of their condition, 67.3% (95%CI 64.8–69.9%, 878/1305) took treatment, and 49.0% (95%CI 46.3–51.7%, 640/1305) were controlled. Among participants with diabetes, 48.4% (95%CI 42.0–55.0%, 111/229) were aware of their condition, 55.8% (95%CI 49.5–62.3%, 128/229) took treatment, and 41.5% (95%CI 35.1–47.9%, 95/229) were controlled. For hypertension, women had higher odds of being on treatment (adjusted odds ratio (aOR) 2.54, 95% CI 1.78–3.61) and controlled (aOR 2.44, 95%CI 1.76–3.37) than men. Participants from urban areas had lower odds of being on treatment (aOR 0.63, 95% CI 0.44–0.90) or being controlled (aOR 0.63, 95% CI 0.46–0.85). Considerable gaps along the hypertension and diabetes care cascades in Lesotho indicate that access and quality of care for these conditions are insufficient to ensure adequate long-term health outcomes.

manuscript on zenodo.org. It can be accessed under: https://doi.org/10.5281/zenodo.8366615

**Funding:** This study is part of the ComBaCaL project which is funded by the TRANSFORM grant of the Swiss Agency for Development and Cooperation (project number 7F-10345.01.01) and a grant by the World Diabetes Foundation (WDF-1778). NDL receives his salary from the Swiss National Science Foundation (SNSF Eccellenza PCEFP3_181355). AA's salary is funded through a career grant of the University of Basel (Junior Research Fund) and EU RESPONSE. EF receives his salary from the European Union's Horizon 2020 research and innovation programme under the Marie Skłodowska-Curie grant agreement (No 801076), through the SSPH+ Global PhD Fellowship Programme in Public Health Sciences (GlobalP3HS). The funders had no role in study design, data collection, analysis, decision to publish, or preparation of the manuscript.

**Competing interests:** The authors have declared that no competing interests exist.

## Introduction

Hypertension and diabetes mellitus are risk factors linked to cardiovascular diseases (CVD) and paramount causes of mortality globally [1–3]. The sub-Saharan African region, including Lesotho, records rising rates of hypertension and diabetes and increasing CVD-related morbidity and mortality [4–8]. Despite this, these conditions remain largely under-diagnosed, and access to high-quality treatment is inadequate [9–12].

Hypertension and diabetes care cascades describe a series of sequential steps in a patient's journey to reach treatment control. Various steps are evaluated: screening and diagnosis of the condition including disease awareness, treatment initiation, long-term retention in care, and treatment control. Quality of care of a health program can be assessed by monitoring each step (i.e., minimum drop-out, maximum rate of treatment control). Countries in sub-Saharan Africa have begun promoting care cascades as a regular monitoring tool to enhance hypertension and diabetes treatment outcomes [13–15].

In Lesotho health services have traditionally been geared towards the control of infectious diseases and infant and maternal mortality [16–18]. Recent estimates from Lesotho suggest a 21.6% prevalence of hypertension and a 5.3% prevalence of diabetes in adults of 18 years and older [19]. However, recent data about awareness, treatment and control of hypertension and diabetes are scarce [20] and are not routinely used to guide programmatic interventions. In the context of a broad population-based survey that measured a wide range of non-communicable diseases in two of ten districts of the country, this study aimed to measure the hypertension and diabetes care cascade, and to identify factors associated with being on treatment and reaching treatment targets.

## Methods

### Study setting

This study is part of a population-based, cross-sectional prevalence survey conducted in the districts of Butha-Buthe and Mokhotlong, in Lesotho, from 1st November 2021 to 31st August 2022. The study was nested within the Community-based Chronic Care Lesotho Project (ComBaCaL; www.combacal.org). The first part of the project included a large population-based, cross-sectional prevalence survey assessing different chronic diseases and cardiovascular disease risk factors among individuals living in the project implementation districts in northern Lesotho. More information about the survey including the prevalence data have been published elsewhere [19]. In this study, we report the full care cascades to complement the published prevalence data.

Lesotho has a population just over 2 million and is classified as a lower-middle-income country [21]. In 2021, the Butha-Buthe district had a total population of approximately 150,000 inhabitants; 35,000 living in Butha-Buthe town, the remaining in often remote villages scattered over an area of 1767km$^2$. The district has one district hospital, one missionary hospital and 12 peripheral clinics (six public, three missionary, and three private-for-profit facilities). The Mokhotlong district has approximately 100,000 inhabitants, of which 30,000 live in Mokhotlong town, with the remaining living in villages over an area of 4075km$^2$. The district has one hospital, and nine rural public clinics [22].

### Survey design and field procedures

We randomly selected 120 population clusters, 60 in (peri-)urban and 60 in rural areas. We considered these clusters as primary sampling units and household members as secondary sampling units. Households were eligible if the household head, or an adult representative,

gave verbal consent. Household members were randomly selected using an algorithm based on age, sex and settlement (rural vs urban), programmed in the Open Data Kit data collection software [23]. Selected household members were eligible if they provided written informed consent. This manuscript is a secondary analysis of the main survey results that described the hypertension and diabetes mellitus prevalence estimates. For this manuscript, we only included participants aged 18 years or older.

Participants received oral and written study information in the local language (Sesotho) and gave written informed consent. In case of illiteracy, participants signed with a thumbprint and a witness co-signed the form. Interviews were conducted in Sesotho. We collected information about sociodemographic characteristics (sex, age, marital status, level of education completed, employment), past medical history (self-reported, or described in the patient health booklet) and household wealth. The household wealth was computed using the Demographic and Health Survey (DHS) Program wealth index questions for Lesotho. The DHS wealth index is calculated based on a questionnaire that assesses housing construction characteristics, household assets and utility services, including country-specific assets that are viewed as indicators of economic status [24, 25].

Blood pressure (BP) was measured using the validated Watch BP Office ABI device [26] with participants sitting after resting for 15 minutes, and automatically repeated two more times at 2-minute intervals each. The mean of the last two measurements was used to calculate the final systolic blood pressure (SBP) and diastolic blood pressure (DBP) results. Due to the conduct of several point-of-care tests in parallel, venous blood was collected in EDTA tubes. Random blood glucose (RBG) was measured using the Accu-chek Active testing platform [27]. For participants with RBG $\geq$ 5.6 mmol/L, a glycosylated hemoglobin (HbA1c) measurement was performed using either the Jana Care Aina station or the A1CNow+ Professional system.

## Study population and sample size

Following our survey protocol for the prevalence estimates [19], we included participants who were found to have hypertension, defined as SBP $\geq$140 mmHg and/or DBP $\geq$90 mmHg and/or who reported to be taking antihypertensive medication (self-reported or written in their health booklet). Regarding diabetes, we included participants with confirmed diabetes, defined as RBG $\geq$5.6 mmol/l and HbA1c $\geq$6.5%, or RBG $\geq$11.1 mmol/l in the absence of an HbA1c measurement, or being on antidiabetic treatment. We used HbA1c in addition to establish the diagnosis, because the remoteness of the survey made it difficult to sample two blood sugar measurements on two different days and many participants encountered might not fulfil the criteria for fasting blood glucose. HbA1c has been recommended as alternative to fasting blood glucose for diagnosis of diabetes with high specificity in community settings [28, 29]. The sample size for this study was determined by the sample size of the overarching survey that aimed at obtaining prevalence estimates and published elsewhere [19].

## Hypertension and diabetes care cascades

For both the hypertension and diabetes cascades, we assessed the following three steps: being aware of the condition, being on treatment for the condition, and being medically controlled for the condition, i.e., reaching treatment targets. A participant was categorized as "being aware" of their condition if they stated that a health care worker had previously told them they had hypertension and/or diabetes mellitus. "On treatment" was defined as participants stating they were taking medication along with a written proof of anti-hypertensive or anti-diabetic medication prescription, respectively, in their health booklet. Hypertension control was

defined as SBP <140mmHg and DBP <90 mmHg on the day of the survey. Diabetes control was defined as a HbA1c ≤ 8.0% on the day of the survey, as suggest by similar studies [30].

## Statistical analysis

We extracted information for each participant from the main study dataset, including sex, age, settlement, relationship status, level of education, employment, self-reported HIV status, body mass index (BMI), and household wealth.

Participants' characteristics were described using frequency and median with interquartile ranges (IQR). The number of participants diagnosed with hypertension or diabetes was set as the denominator for all other steps of the respective care cascade. For each step of the care cascades, we reported frequencies with Wald 95% confidence intervals (CI). We reported adjusted odds ratios (aOR) with 95% CI derived from logistic regression models to assess the association of patient characteristics with the different steps in the care cascades. Missing data were considered missing completely at random. We used the STROBE cross sectional checklist for reporting our results. Statistical analysis was done using Stata/IC (version 16.0, College Station, Tex: StataCorp LP, 2019).

## Ethics statement

All procedures were carried out in line with the ethical standards laid out in the Declaration of Helsinki. Participants received information on all research procedures in Sesotho and gave written informed consent. Illiterate participants gave consent by thumbprint and a witness co-signed the form. Once the informed consent process was completed, a signed copy of the form was retained by study staff and a copy was given to the participant. This study was reviewed by the Ethics Committee Northwest and Central Switzerland (ID AO_2021–00056) and approved by the National Health Research Ethics Committee in Lesotho (ID139-2021).

## Inclusivity in global research and patient and public involvement

This survey is part of the Community-based chronic care project Lesotho (ComBaCaL; www. combacal.org). It was co-designed together with the ComBaCaL steering committee that includes a community representative, as well as representatives from the Ministry of Health of Lesotho. Additional information regarding the ethical, cultural, and scientific considerations specific to inclusivity in global research is included in the S1 Checklist.

# Results

## Participants characteristics

Enrolment of participants is detailed in Fig 1. In the 120 clusters, the study teams visited 3498 households with 7412 eligible household members and identified 1308 participants with hypertension and 291 participants with diabetes as per survey definitions. We excluded participants with missing data in any of the steps along the care cascades. Therefore, we finally included 1305 and 229 participants with hypertension and diabetes, respectively. Table 1 summarizes their characteristics. Women represented a total of 65.6% (856/1305) and 73.8% (169/229) of participants in the hypertension and the diabetes cascades, respectively. Approximately two-thirds of the participants lived in urban or peri-urban settlements and almost half reported having regular income. Overall, approximately one in five participants reported to live with HIV. Overall, 149 participants had both hypertension and diabetes, i.e. 149/229 (65%) participants with diabetes also had hypertension and 149/1305 (11%) participants with hypertension also had diabetes.

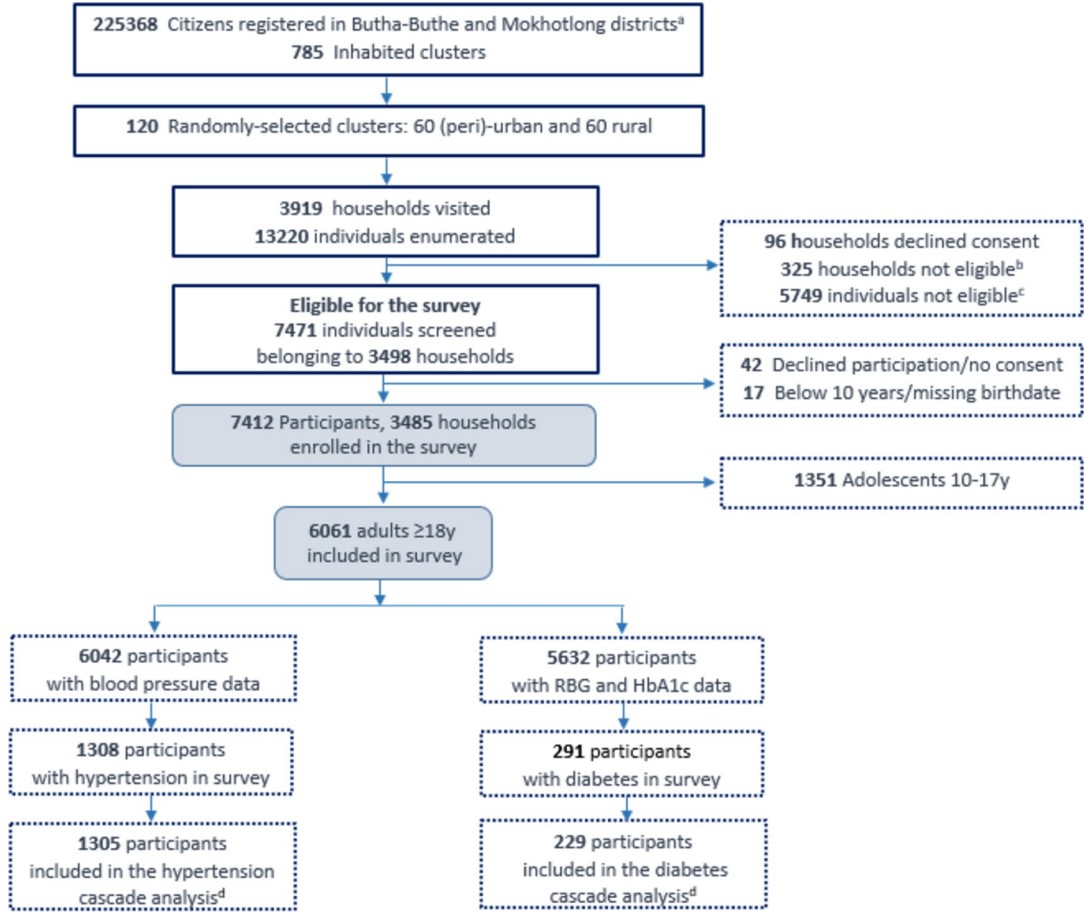

**Fig 1. Study flow.** [a] Lesotho - Subnational Population Statistics-UNFPA: https://data.humdata.org/dataset/cod-ps-lso. [b] Had no participants who were eligible for survey (i.e., participants received a basic package of services: BP and RBG measurements, members who were present were <10 years). [c] Individuals were enumerated but not present at the time of the survey or were not eligible (i.e., <10 years, or those who received a basic to package of services: BP and RBG measurements). [d] Individuals included had information in all four steps of the care cascades. Acronyms: RBG: random blood glucose; HbA1c: glycosylated hemoglobin.

## Care cascades for hypertension and diabetes

Fig 2 shows the care cascades for hypertension and diabetes. Among the participants with hypertension, 69.7% (95%CI 67.2–72.2%, 909/1305) were aware of their condition, 67.3% (95%CI 64.8–69.9%, 878/1305) took medical treatment, and 49.0% (95%CI 46.3–51.7%, 640/1305) had controlled BP on the day of the survey. Among the participants with diabetes, 48.4% (95%CI 42.0–55.0%, 111/229) were previously diagnosed, 55.8% (95%CI 49.5–62.3%, 128/229) took medical treatment, and 41.5% (95%CI 35.1–47.9%, 95/229) were controlled on the day of the survey. When analyzing the care cascades as a proportion of the prior step, a total of 69.7%, (95%CI 67.2–72.1%, 909/1305) participants with hypertension were aware of their condition, out of these, 96.6% (95%CI 95.4–97.8%, 878/909) were on treatment, and 72.9% (95%CI 71.0–76.8%, 640/878) had controlled BP. Among those with diabetes, 48.5% (95%CI 42.0–54.9%, 111/229) were aware of their condition. A total of 128 participants reported being on anti-diabetic treatment, and of those, 74.2%, (95%CI 66.6–81.8%, 95/128) were controlled.

**Table 1. Baseline characteristics of study participants in hypertension and diabetes care cascades.**

| | | Hypertension cascade n (%) | Diabetes cascade n (%) |
|---|---|---|---|
| **Total participants** | | **1305 (100)** | **229 (100)** |
| **Sex** | Female | 856 (65.6) | 169 (73.8) |
| | Male | 449 (34.4) | 60 (26.2) |
| **Age** | <30 years | 69 (5.3) | 19 (8.3) |
| | 30–59 years | 537 (41.1) | 93 (40.6) |
| | ≥ 60 years | 699 (53.6) | 117 (51.1) |
| **Settlement** | Urban/peri-urban | 797 (61.1) | 159 (69.4) |
| | Rural | 508 (38.9) | 70 (30.6) |
| **Relationship status** | Married/in a relationship | 704 (53.9) | 128 (55.9) |
| | Single/divorced/widowed | 601 (46.1) | 101 (44.1) |
| **Education level** | No schooling | 148 (11.3) | 10 (4.3) |
| | Primary school | 761 (58.3) | 135 (58.9) |
| | Secondary | 320 (24.5) | 62 (27.1) |
| | Tertiary | 75 (5.8) | 21 (9.3) |
| | Missing | 1 (0.1) | 1 (0.4) |
| **Employment status** | (Self-) Employment/regular income | 583 (44.7) | 106 (46.3) |
| | No employment/no regular income | 722 (55.3) | 123 (53.7) |
| HIV status* | Positive | 227 (17.4) | 45 (19.6) |
| | Negative ≤12 m ago | 458 (35.1) | 85 (37.1) |
| | Negative >12 m ago | 291 (22.3) | 51 (22.3) |
| | Missing/unknown | 329 (25.2) | 48 (21.0) |
| **Diabetes status** | Also diagnosed with diabetes | 149 (11.4) | NA |
| | Not diagnosed with diabetes | 1156 (88.6) | NA |
| **Hypertension status** | Also diagnosed with hypertension | NA | 149 (65.1) |
| | Not diagnosed with hypertension | NA | 80 (34.9) |

*Self-reported or recorded in participants' health card

## Factors associated with hypertension and diabetes treatment and control

Table 2 provides the estimates from univariate and multivariable logistic regression analyses. For hypertension, women had higher odds of being on treatment (aOR 2.54, 95% CI 1.78–3.61) and achieving BP control (aOR 2.44, 95% CI 1.76–3.37) than men. Participants from urban areas had lower odds of being on treatment (OR 0.63, 95% CI 0.44–0.90) or having controlled BP (aOR 0.63, 95% CI 0.43–0.85), than participants living in rural areas. A higher BMI was associated with higher odds of being on treatment (aOR 1.49, 95% CI 1.07–2.07), but not with having controlled BP. Overall, compared to participants less than 30 years old, older participants had higher odds of being on treatment and having controlled BP. We did not find significant associations with living with HIV, nor with household wealth, education, or employment. For diabetes, the oldest age group (60 years and older), had highest odds of being on treatment (OR 3.44, 95% CI 1.03–11.49). None of the covariates included in our model showed a significant association with diabetes control.

## Discussion

Our findings indicate that the gaps along the hypertension and diabetes care cascades in Lesotho are lower than the regional average [30–32]. However, taking the 80-80-80 global targets

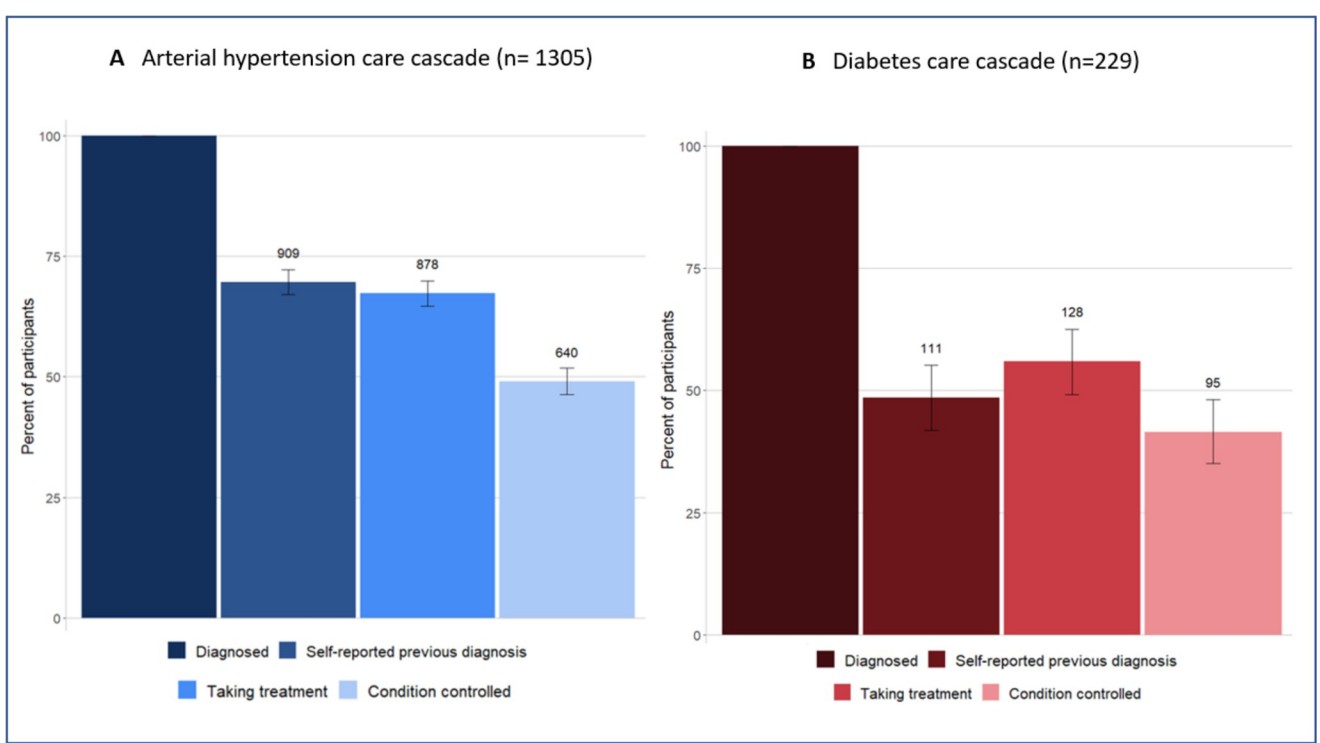

**Fig 2.** Care cascades for hypertension (A) and diabetes (B), showing the percentage, with 95% confidence intervals, and number of participants in each step of the cascade.

for the diabetes and hypertension care cascades as a reference [33, 34], our results still show insufficient rates of awareness, treatment, and control for both conditions. With regards to hypertension, one in three adults did not know they had hypertension, nor had initiated treatment, and only half of all the adults with hypertension had adequate BP control. The diabetes cascade shows even larger gaps across all the steps. Only half of the adults that screened positive for diabetes had a previous diagnosis or had initiated treatment and glycemic control was achieved by less than half of the participants.

Curbing the morbidity and mortality of hypertension and diabetes in sub-Saharan Africa will require identification of all those who remain undiagnosed, untreated, and uncontrolled [31]. Routine monitoring of care cascades is a fundamental tool to achieve these objectives in health programs. Once established, CVDs programs must attain high rates of BP and glycemic control and integrate comprehensive CVDs risks management. Health systems strengthening approaches are needed to ensure no cascade gaps, including a functional supply chain for drugs and diagnostic commodities, capacity building for task shifted care, and availability of routine BP and HbA1c measurements (including point-of-care devices). Much can be learned from the HIV and tuberculosis programmes [35–39].

Our study shows that the performance of both cascades, although better than the regional average [31], remains suboptimal in Lesotho. There are significant gaps across all steps, and BP and glycemic control are insufficient to substantially decrease CVD risk in this population. Adults with hypertension appeared to have a higher level of awareness (69.6%, 95%CI 67.2–72.2%), than those with diabetes (48.4%, 95%CI 42.0–55.0%), indicating a more frequent routine measurement of BP than glycemia.

**Table 2. Univariate and multivariable regression analysis for previous treatment and condition control in participants of the hypertension and diabetes care cascades.**

| | Hypertension (n = 1305) | | | | Diabetes (n = 229) | | | |
|---|---|---|---|---|---|---|---|---|
| | On treatment | | Blood pressure controlled | | On treatment | | Diabetes controlled | |
| | OR (95%CI) | aOR (95%CI)$^§$ | OR (95%CI) | aOR (95%CI)$^§$ | OR (95%CI) | aOR (95%CI)$^Υ$ | OR (95%CI) | aOR (95%CI)$^Υ$ |
| **Sex** | | | | | | | | |
| Male | Reference | Reference | Reference | Reference | Reference | Reference | Reference | Reference |
| Female | 2.97 (2.33–3.78) | **2.54 (1.78–3.61)** | 2.27 (1.80–2.87) | **2.44 (1.76–3.37)** | 1.51 (0.83–2.73) | 1.90 (0.93–3.89) | 1.19 (0.65–2.18) | 1.25 (0.61–2.54) |
| **Age group** | | | | | | | | |
| <30y | Reference | Reference | Reference | Reference | Reference | Reference | Reference | Reference |
| 30-59y | 2.42 (1.44–4.07) | **2.45 (1.28–4.71)** | 1.98 (1.14–3.42) | **2.44 (1.26–4.75)** | 3.87 (1.29–11.65) | 2.49 (0.71–8.65) | 1.50 (0.52–4.29) | 1.44 (0.43–4.82) |
| ≥ 60y | 6.07 (3.60–10.24) | **7.09 (3.61–13.92)** | 2.92 (1.70–5.01) | **3.09 (1.58–6.06)** | 4.02 (1.36–11.91) | **3.44 (1.03–11.49)** | 1.67 (0.59–4.71) | 1.77 (0.56–5.61) |
| **Education** | | | | | | | | |
| No/Primary | Reference | Reference | Reference | Reference | Reference | Reference | Reference | Reference |
| Secondary/Tertiary | 0.71 (0.56–0.92)$^R$ | 1.33 (0.93–1.90) | 0.70 (0.55–0.89)$^R$ | 1.01 (0.80–1.52) | 0.95 (0.55–0.64)$^μ$ | 1.04 (0.54–2.01) | 0.95 (0.55–1.66)$^μ$ | 1.23 (0.64–3.37) |
| **Stable relation** | | | | | | | | |
| No | Reference | Reference | Reference | Reference | Reference | Reference | Reference | Reference |
| Yes | 0.94 (0.74–1.18) | 1.39 (0.99–1.93) | 0.89 (0.72–1.11) | 1.21 (0.90–1.62) | 1.84 (1.08–3.12) | **1.98 (1.06–3.71)** | 1.06 (0.63–1.81) | 1.07 (0.58–1.99) |
| **Employment/Regular income** | | | | | | | | |
| No | Reference | Reference | Reference | Reference | Reference | Reference | Reference | Reference |
| Yes | 1.84 (1.45–2.32) | 1.02 (0.73–1.42) | 1.49 (1.19–1.85) | 0.99 (0.74–1.33) | 1.10 (0.65–1.85) | 0.83 (0.44–1.54) | 0.93 (0.55–1.57) | 0–79 (0.36–1.41) |
| **Settlement** | | | | | | | | |
| Rural | Reference | Reference | Reference | Reference | Reference | Reference | Reference | Reference |
| Urban | 0.60 (0.46–0.76) | **0.63 (0.44–0.90)** | 0.59 (0.47–0.74) | **0.63 (0.46–0.85)** | 0.78 (0.55–1.39) | 0.76 (0.40–1.43) | 0.60 (0.34–1.07) | 0.63 (0.33–1.19) |
| **BMI ≥ 25 Kg/m$^2$** | | | | | | | | |
| No | Reference | Reference | Reference | Reference | Reference | Reference | Reference | Reference |
| Yes | 2.24 (1.76–2.84)$^ß$ | **1.49 (1.07–2.07)** | 1.31 (1.04–1.64)$^ß$ | 0.87 (0.65–1.18) | 1.50 (0.84–2.68)$^#$ | 0.97 (0.49–1.92) | 0.83 (0.46–1.49)$^#$ | 0.72 (0.36–1.41) |
| **Known HIV-positive$^α$** | | | | | | | | |
| No | Reference | Reference | Reference | Reference | Reference | Reference | Reference | Reference |
| Yes | 0.98 (0.71–1.36)$^€$ | 1.17 (0.81–1.69) | 0.93 (0.69–1.25)$^{€tb2e}$ | 0.92 (0.67–1.28) | 0.70 (0.37–1.35) | 0.58 (0.29–1.16) | 0.92 (0.47–1.80) | 0.86 (0.42–1.75) |
| **Household wealth** | | | | | | | | |
| Q1 | 0.95 (0.64–1.40)$^¥$ | 1.19 (0.69–2.05) | 1.16 (0.80–1.68)$^¥$ | 1.50 (1.76–3.37) | 0.74 (0.29–1.99) | 0.71 (0.24–2.06) | 0.63 (0.23–1.72) | 0.55 (0.19–1.60) |
| Q2 | 0.94 (0.63–1.41) | 0.91 (0.53–1.56) | 1.12 (0.76–1.63) | 1.15 (0.94–2.38) | 0.65 (0.23–1.90) | 1.02 (0.33–3.10) | 0.32 (0.11–0.92) | 0.30 (0.10–0.92) |
| Q3 | Reference | Reference | Reference | Reference | Reference | Reference | Reference | Reference |
| Q4 | 0.96 (0.66–1.38) | 0.81 (0.50–1.34) | 0.97 (0.69–1.37) | 1.07 (0.69–1.28) | 0.93 (0.46–2.43) | 1.78 (0.68–4.64) | 0.74 (0.31–1.78) | 0.65 (0.26–1.63) |

(Continued)

**Table 2.** (Continued)

| | Hypertension (n = 1305) | | | | Diabetes (n = 229) | | | |
| | On treatment | | Blood pressure controlled | | On treatment | | Diabetes controlled | |
| | OR (95%CI) | aOR (95%CI)§ | OR (95%CI) | aOR (95%CI)§ | OR (95%CI) | aOR (95%CI)¥ | OR (95%CI) | aOR (95%CI)¥ |
|---|---|---|---|---|---|---|---|---|
| Q5 | 1.04 (0.72–1.50) | 1.07 (0.65–1.76) | 0.99 (0.70–1.39) | 1.28 (0.83–1.98) | 0.89 (0.36–2.20) | 1.48 (0.61–3.58) | 0.48 (0.21–1.10) | 0.47 (0.19–1.15) |

ℝ Total number of observations 1304.

ß Total number of observations 1262.

α Self-reported.

€ Total number of observations 976.

¥ Total number of observations 1299.

§ Multivariate model fitted on a reduced dataset of 941 observations (participants with missing covariate information were dropped).

μ Total number of observations 228.

# Total number of observations 223.

ϒ Multivariate model fitted on a reduced dataset of 222 observations (participants with missing covariate information were dropped).

In 2014, the Lesotho Demographic and Health Survey [40] reported that 68.5% of the participants who were found to have hypertension were on treatment, and 70.0% of participants who were found to have diabetes had started treatment, representing similar findings to this study. The 2012 National Health and Nutrition Examination Survey [15] in neighboring South Africa reported that half of the adults who screened positive for hypertension were undiagnosed, 78% of those with hypertension were not on treatment and, overall, only 9% of participants with hypertension had achieved the threshold of treatment control. A recent analysis of the continuum of care in Tanzania [41] revealed important gaps in both cascades. In this setting only 21% and 11% of participants with hypertension were on treatment and had achieved BP controlled, respectively, whereas 66% and 48% of participants with diabetes had started treatment and had achieved glycemic control, respectively. The authors suggested that these gaps were heavily influenced by out-of-pocket expenses, i.e., patient fees for services, as indicated by studies from South Africa [42, 43] and multi-country analyses [32]. Our study found no significant association between household wealth and treatment initiation or control. In Lesotho, services for hypertension and diabetes are predominantly free in health centers, and patients are only required to pay a minimal fee when attending hospital services. Our results may be an indication of less inequality of chronic care services in Lesotho, providing similar opportunities to initiate and maintain treatment regardless of wealth. However, this study was not specifically designed to address this question and further research is needed to explore this association.

We found disparities in treatment and control between hypertension and diabetes with worse outcomes across the diabetes cascade. This may be attributed to different factors. For example, clinics may lack the necessary equipment for regular glucose monitoring, whereas basic tools for BP measurement are more available, or healthcare professionals may be more aware and knowledgeable about hypertension and its complications than diabetes.

We found that participants with hypertension who were male and below 30 years, had lower odds of taking treatment or being controlled than women, elder age groups or those in rural areas. Such findings are consistently reported in other similar studies [41, 44–46]. However, contrary to previous studies which have suggested that awareness, and control tend to be higher in urban areas [47, 48], our study found that adults living in urban areas had lower

odds of being on hypertension treatment or achieving blood pressure control. This was not observed for diabetes. We hypothesize that this could be due to longer waiting times, rushed consultations, and inadequate follow-up at urban health facilities. Additionally, urban residents may be more mobile than their rural counterparts. However, these are speculations and need further research.

A growing body of evidence fosters the idea that HIV treatment programs can successfully integrate, and thus strengthen, care for CVDs in settings with a HIV prevalence such as Lesotho [35, 49–51] For example, a recent study reviewing hypertension and diabetes outcomes along the HIV care cascade in rural South Africa found that participants living with HIV had lower systolic BP and blood glucose than participants not living with the virus [50]. We did not find a significant association between participants living with HIV and better outcomes across the hypertension and diabetes care cascades.

Our study has several limitations. The information for HIV status, and the first step (awareness) in the cascades are collected from self-reported information [52, 53]. This is the reason why in the diabetes care cascade (Fig 2B) the proportion of participants being aware or recalling a previous diagnosis is lower than those who reported being on diabetes treatment. Thus, this data needs to be interpreted with caution and we decided not to include this step in the multivariable analyses. It is indeed possible that people with diabetes and taking antidiabetic treatment are unaware of their diagnosis, for example due to delegated care or polypharmacy due to multimorbidity. Second, as this was a cross-sectional household-based survey, hypertension diagnosis and control relied on single-day measurements. However, this is the standard approach recommended by WHO for household-based surveys, such as STEPwise approach to NCD risk factor surveillance surveys, in these settings [8]. Third, from approximately one third of the participants we did not have information on their HIV status. This was due to procedural changes during the survey conduct, and there is no reason to believe the survey population was different after the procedural change than before, thus we consider this data being missing completely at random.

## Conclusion

The cascades of care for hypertension and diabetes in Lesotho have considerable gaps, indicating that access and quality of services remain insufficient to ensure adequate health outcomes. This study provides a baseline for future routine monitoring of these indicators across age and sex groups to support the progress to achieving programmatic targets in this setting.

## Supporting information

**S1 Checklist. Inclusivity in global research.**
(PDF)

**S2 Checklist. STROBE checklist.**
(PDF)

## Acknowledgments

The authors would like to express their gratitude to all the involved staff at the Ministry of Health in Lesotho, the SolidarMed team in Lesotho, to the extended ComBaCaL survey team, and to all the participants in this study.

## Author Contributions

**Conceptualization:** Lucia González Fernández, Emmanuel Firima, Ravi Gupta, Makhebe Khomolishoele, Manthabiseng Molulela, Thabo Ishmael Lejone, Niklaus Daniel Labhardt, Alain Amstutz.

**Data curation:** Lucia González Fernández, Tristan Lee, Frédérique Chammartin.

**Formal analysis:** Lucia González Fernández, Tristan Lee, Frédérique Chammartin.

**Funding acquisition:** Niklaus Daniel Labhardt, Alain Amstutz.

**Investigation:** Lucia González Fernández, Emmanuel Firima, Ravi Gupta, Mamoronts'ane Pauline Sematle, Makhebe Khomolishoele, Manthabiseng Molulela, Matumaole Bane, Mosa Tlahali, Stephen McCrosky, Tristan Lee, Frédérique Chammartin, Eleonora Seelig, Felix Gerber, Thabo Ishmael Lejone, Irene Ayakaka, Niklaus Daniel Labhardt, Alain Amstutz.

**Methodology:** Lucia González Fernández, Emmanuel Firima, Ravi Gupta, Mamoronts'ane Pauline Sematle, Makhebe Khomolishoele, Manthabiseng Molulela, Matumaole Bane, Mosa Tlahali, Stephen McCrosky, Tristan Lee, Frédérique Chammartin, Eleonora Seelig, Felix Gerber, Thabo Ishmael Lejone, Irene Ayakaka, Niklaus Daniel Labhardt, Alain Amstutz.

**Project administration:** Ravi Gupta, Mamoronts'ane Pauline Sematle, Makhebe Khomolishoele, Manthabiseng Molulela, Matumaole Bane, Stephen McCrosky, Felix Gerber, Thabo Ishmael Lejone, Irene Ayakaka, Niklaus Daniel Labhardt, Alain Amstutz.

**Resources:** Stephen McCrosky, Irene Ayakaka, Niklaus Daniel Labhardt, Alain Amstutz.

**Supervision:** Niklaus Daniel Labhardt, Alain Amstutz.

**Validation:** Alain Amstutz.

**Writing – original draft:** Lucia González Fernández, Alain Amstutz.

**Writing – review & editing:** Emmanuel Firima, Ravi Gupta, Mamoronts'ane Pauline Sematle, Makhebe Khomolishoele, Manthabiseng Molulela, Matumaole Bane, Mosa Tlahali, Stephen McCrosky, Tristan Lee, Frédérique Chammartin, Eleonora Seelig, Felix Gerber, Thabo Ishmael Lejone, Irene Ayakaka, Niklaus Daniel Labhardt, Alain Amstutz.

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
