## [Decision Letter · Decision Letter 0]

20 May 2024

PGPH-D-24-00714

Awareness, treatment, and control among adults living with arterial hypertension or diabetes mellitus in two rural districts in Lesotho

Dear Dr. Amstutz,

Thank you for submitting your manuscript to PLOS Global Public Health. After careful consideration, we feel that it has merit but does not fully meet PLOS Global Public Health’s publication criteria as it currently stands. Therefore, we invite you to submit a revised version of the manuscript that addresses the points raised during the review process.

We look forward to receiving your revised manuscript.

Kind regards,

Ikechi G Okpechi

Academic Editor

Journal Requirements:

2. We do not publish any copyright or trademark symbols that usually accompany proprietary names, eg  ©, ®, ™  (e.g. next to drug or reagent names). Please remove all instances of trademark/copyright symbols throughout the text, including ® on page 18.

Additional Editor Comments (if provided):

Reviewers' comments:

Reviewer's Responses to Questions

**Comments to the Author**

1. Does this manuscript meet PLOS Global Public Health’s publication criteria? Is the manuscript technically sound, and do the data support the conclusions? The manuscript must describe methodologically and ethically rigorous research with conclusions that are appropriately drawn based on the data presented.

Reviewer #1: Yes

Reviewer #2: Yes

2. Has the statistical analysis been performed appropriately and rigorously?

Reviewer #1: Yes

Reviewer #2: No

3. Have the authors made all data underlying the findings in their manuscript fully available (please refer to the Data Availability Statement at the start of the manuscript PDF file)?

Reviewer #1: No

Reviewer #2: Yes

4. Is the manuscript presented in an intelligible fashion and written in standard English?

Reviewer #1: Yes

Reviewer #2: No

5. Review Comments to the Author

Reviewer #1: This population-based survey investigated the levels of awareness, treatment, and control regarding hypertension and diabetes in Lesotho. The study is particularly intriguing due to its identification and illustration of notable gaps within the hypertension and diabetes care cascade. However, I have some comments to add.

Major comments

1. Study setting: What factors influenced the choice of Butha-Buthe and Mokhotlong for the survey? Is there any other important characteristics beyond population size and hospital availability. Were considerations such as the prevalence of hypertension and diabetes, healthcare accessibility, demographic composition, existing healthcare infrastructure and resources, availability of trained professionals, and community participation, taken into account?

2. Study participants: What was the number of participants who had both hypertension and diabetes? Additionally, could you describe the characteristics of the cascade within this subgroup?

3. Discussion: What factors do the authors believe contribute to the disparity in awareness, treatment, and control between hypertension and diabetes?

4. Discussion: The findings indicated a lower number of individuals aware of their condition than those receiving treatment, with the authors attributing this difference to self-reporting. However, there may be additional factors influencing this, such as participants who are on medication but unaware that they have diabetes. Could this be linked to the health literacy or educational level of the population, or the patient education during treatment care?

5. Discussion: What factors could explain why individuals in urban or peri-urban areas had lower odds of awareness and control compared to those in rural areas? Typically, urban areas are expected to have better access to diagnosis and treatment, so why did this trend differ in the study results and contradict from previous articles?

6. Discussion: Are there existing gaps in Lesotho's NCD strategies, and how might this study influence policy changes? Please elaborate more on how the findings could be implemented in practice?

Minor comments

1. In page 4, line 113, please also cite the main survey article here.

2. The percentage for the age group 30-59 in the diabetes cascade within Table 1 should be corrected to 40.6%.

Reviewer #2: Thank you for the opportunity to review this paper

General comments

1. Will like to congratulate the authors for the work done and the willingness to get their study published. Well done!

2. Was keen to know why authors are not interested in reporting prevalence based on their chosen methodology for which they could have increased the sample size especially for the participants with diabetes

3. I have a fundamental challenge with the definition of diabetes and the definition of control of diabetes in the study

4. Generally, language and grammatical error should be checked.

5. Not clear on how sample sizes were arrived at from the methodolgy. It should be clearly stated

6. Discussion needs more revision to make the study meaningful

Introduction

1. “Lesotho has a population just over 2 million and is classified as a lower-middle-income country16.’ This statement should be part of the methods instead as readers get an idea of the study setting.

2. If authors prefer to describe hypertension as ‘arterial hypertension” then its preferable to be more complete at systemic arterial hypertension or just keep as hypertension might even suffice.

3. Might be helpful for readers to know the prevalence of hypertension and diabetes in Lesotho or as well as the burden of disease to make stronger case for the study.

Methods

1. The definition of diabetes with random blood glucose of >5.6 and Hba1c > 6.5% may be low and the definition of HbA1c of less than 8.0% for diabetes controlled is quite high. What is the source of this definition? It should be clearly stated and referenced. What about those with previous HbA1c of less than 8.0 already at diagnosis. Will they be considered good control if on medications?

2. Participants characteristics - Enrolment of participants is detailed in figure 1. “In the 120 clusters, the study teams visited 3498 households with 7412 eligible household members and identified 1308 participants with arterial hypertension and 291 participants with diabetes as per survey definitions.’ If sample size was well calculated enough prevalence of DM and HPT in Lesotho could be reported. Can authors work on this?

3. There is supposed to be a figure of study flow but can’t be seen on page 9 but its legend. Authors should please rectify.

4. What informed the varying sample size, the authors should explain vividly in the methods

Results

1. Study is reporting very high awareness rate and high control rates for both DM and hypertension

2. Authors should have studied a few more variables like family history of the convictions. Risk factors such as smoking and alcohol intake, sedentary lifestyle etc. as well as complication of the diabetes and hypertension – this made the discussion quite hollow considering the good data collected

Discussion

The discussion needs a lot of tightening considering the study conducted. Major revision required with the discussion. Some few examples as a guide.

1. “Our results show insufficient rates of awareness, treatment, and control for both conditions. With regards to hypertension, one in three adults did not know they had hypertension, nor had initiated treatment, and only half of all the adults with hypertension had adequate BP control.”

Considering the relatively good awareness and control rate I think author should begin the discussion with that opening statement that compares to normal referenced estimates.

2. “Our study found no significant association between household wealth and awareness, treatment initiation, or control. How does this compare to other studies? What could be the possible reasons? That should be discussed thoroughly.

3. In our study, we did not find a significant association between participants living with HIV and outcomes across the hypertension or diabetes care cascades.

Wondering why authors are rather interested in HIV and its association with DM and hypertension? Authors should focus more on the traditional risk factors or modifiable factors for diabetes and hypertension.

6. PLOS authors have the option to publish the peer review history of their article (what does this mean?). If published, this will include your full peer review and any attached files.

**Do you want your identity to be public for this peer review?** For information about this choice, including consent withdrawal, please see our Privacy Policy.

Reviewer #1: No

Reviewer #2: No

---

## [Decision Letter · Decision Letter 1]

28 Aug 2024

Awareness, treatment, and control among adults living with arterial hypertension or diabetes mellitus in two rural districts in Lesotho

PGPH-D-24-00714R1

Dear Dr Amstutz,

We are pleased to inform you that your manuscript 'Awareness, treatment, and control among adults living with arterial hypertension or diabetes mellitus in two rural districts in Lesotho' has been provisionally accepted for publication in PLOS Global Public Health.

Best regards,

Ikechi G Okpechi

Academic Editor

Reviewer Comments (if any, and for reference):

Reviewer's Responses to Questions

**Comments to the Author**

1. If the authors have adequately addressed your comments raised in a previous round of review and you feel that this manuscript is now acceptable for publication, you may indicate that here to bypass the “Comments to the Author” section, enter your conflict of interest statement in the “Confidential to Editor” section, and submit your "Accept" recommendation.

Reviewer #1: All comments have been addressed

Reviewer #2: All comments have been addressed

2. Does this manuscript meet PLOS Global Public Health’s publication criteria? Is the manuscript technically sound, and do the data support the conclusions? The manuscript must describe methodologically and ethically rigorous research with conclusions that are appropriately drawn based on the data presented.

Reviewer #1: Yes

Reviewer #2: Yes

3. Has the statistical analysis been performed appropriately and rigorously?

Reviewer #1: Yes

Reviewer #2: Yes

4. Have the authors made all data underlying the findings in their manuscript fully available (please refer to the Data Availability Statement at the start of the manuscript PDF file)?

Reviewer #1: Yes

Reviewer #2: Yes

5. Is the manuscript presented in an intelligible fashion and written in standard English?

Reviewer #1: Yes

Reviewer #2: Yes

6. Review Comments to the Author

Reviewer #1: The author has addressed most of the comments and suggestions provided. I believe the manuscript is now suitable for acceptance.

Reviewer #2: Congratulations to the authors!

7. PLOS authors have the option to publish the peer review history of their article (what does this mean?). If published, this will include your full peer review and any attached files.

**Do you want your identity to be public for this peer review?** For information about this choice, including consent withdrawal, please see our Privacy Policy.

Reviewer #1: No

Reviewer #2: No
